# MixMin: Finding Data Mixtures via Convex Minimization

**Anvith Thudi** [1 2]  **Evianne Rovers** [3 4]  **Yangjun Ruan** [1 2]  **Tristan Thrush** [5]  **Chris J. Maddison** [1 2]

## Abstract

Modern machine learning pipelines are increasingly combining and mixing data from diverse and disparate sources, e.g., pre-training large language models. Yet, finding the optimal data mixture is a challenging and open problem. We formalize this data mixing problem as a bi-level objective: the best mixture is the one that would lead to the best model for a downstream objective. Unfortunately, this objective is generally intractable. In this paper, we make the observation that the bi-level data mixing objective becomes convex as our model class becomes larger. We develop and study a gradient-based approach for optimizing this convex objective, which we call MixMin, and test it on language modeling and chemistry tasks. MixMin was the only method that uniformly improved the data mixture in all our experiments. With MixMin, we improved the data mixture using less than $0.2\%$ additional compute for a pythia-$410M$ model trained on $8.2B$ tokens, resulting between 1-5% relative improvement to negative log likelihood on PIQA, ARC Easy, SciQ, and OpenWebMath. Crucially, we found that MixMin mixtures for smaller models improved training of larger models, suggesting that MixMin mixtures may be scale-invariant. When mixing bioassay data to train an XGBoost model, we saw improvements to average precision scores of $0.03 - 0.15$.

## 1. Introduction

Recent progress in ML has come from training on vast web-scale data (Dubey et al., 2024; Touvron et al., 2023;

---
[1]Department of Computer Science, University of Toronto, Toronto, Canada [2]Vector Institute, Toronto, Canada [3]Department of Chemistry, University of Toronto, Toronto, Canada [4]Structural Genomics Consortium, Toronto, Canada [5]Department of Computer Science, Stanford University, Palo alto, USA. Correspondence to: Anvith Thudi <anvith.thudi@mail.utoronto.ca>.

*Proceedings of the 42nd International Conference on Machine Learning*, Vancouver, Canada. PMLR 267, 2025. Copyright 2025 by the author(s).

Raffel et al., 2020; Achiam et al., 2023). These models are known to generalize to data-poor downstream tasks, a core motivation behind scaling training data. However, as we increase the amount and diversity of data sources we can train on, a core challenge becomes choosing how to weigh these sources when training. The effectiveness of a model pre-trained on surrogate data is known to be directly related to how similar the surrogate (or "pretraining") data is to the downstream task (target distribution) (Ben-David et al., 2010; Isik et al., 2024; Jain et al., 2024; Pouget et al., 2024).

Formally, finding the best mixture of data sources to train on for a downstream loss poses a bi-level optimization which is hard to optimize. We must find the mixture whose risk minimizer optimizes the downstream loss. This generally only admits expensive zero-order approaches, such as grid search (Dubey et al., 2024; Blakeney et al., 2024). Past work has considered learning to simulate the loss oracle for unseen mixtures (Liu et al., 2024), or leveraging a large pool of existing models (Thrush et al., 2024). Other approaches consider solving data mixing objectives agnostic to a downstream loss, but are known to not consistently improve the baseline of training on all the data with their natural proportions (when evaluated on diverse downstream losses) (Xie et al., 2024; Jiang et al., 2024; Liu et al., 2024; Fan et al., 2023; Held et al., 2025).

In this paper we show that the bi-level optimization for data mixing reduces to a convex minimization as the model classes become larger, which we can solve cheaply with gradient based methods. A key insight is that by restricting the loss functions to cross-entropy (CE) or mean-squared error (MSE), we gain additional structure which simplifies the data mixing objective. Practically, we show we can find the best mixture by first training a (cheap) proxy model for each source, learning the best mixture of their outputs for the target dataset (convex minimization), and then using this weighting to remix the data, see Figure 1.

First, we note that when our models are relatively close to Bayes optimal (the best possible functions), data mixing is equivalent to learning a linear model over the predictions of a fixed set of models (a convex objective). Learning this linear model requires comparatively little data to training modern large models, and is relatively cheap to optimize. Our reduction is specific to CE or MSE, where we leverage

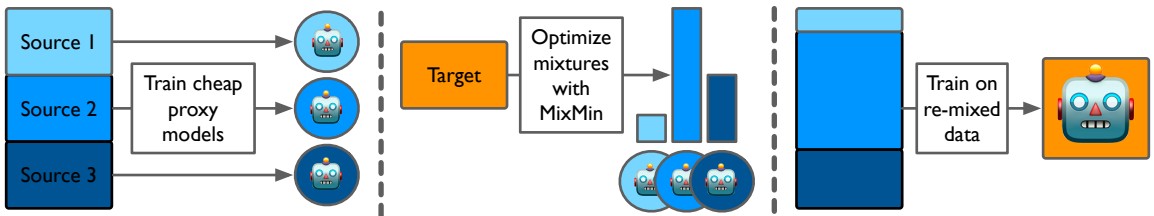

Figure 1: Optimizing MixMin to find the mixture weights requires training a few cheap models for each source and a target dataset. Given the mixture weights we train a more expensive model using the mixture.

the fact the Bayes optimal model for a mixture is the mixture of the Bayes optimal models for each source. However, this reduction does not yet make data mixing amenable to first-order optimization. To evaluate our convex objective's gradient we need the Bayes optimal model for each source.

A key challenge is that the proxy models needed to evaluate this convex objective can be expensive to compute. We find that in practice, we can use cheaply computed models and still have good data mixing performance. We call this approach to optimizing the convex objective *MixMin*. Empirically, we found the performance of MixMin did not significantly degrade between using proxy models computed with $100\%$ the cost of a full training run to $1\%$. In other cases, the MixMin mixture significantly improved over baselines, but the ensemble of proxy models performed much worse than retraining, suggesting they were far from Bayes optimal but did not hinder MixMin performance.

We empirically compared data mixing with MixMin to current baselines on both language modeling and chemistry tasks. For language modeling we considered optimizing the mixture of source domains in SlimPajama (cer, 2023) for PIQA (Bisk et al., 2020), ARC Easy (Clark et al., 2018), SciQ (Welbl et al., 2017), or OpenWebMath (Paster et al., 2023). In all cases we found MixMin improved the baselines for log likelihood, improving or maintaining the gap as we increased the scale of the models from pythia-$160M$ to pythia-$410M$ (Biderman et al., 2023). Moreover, for pythia-$410M$ the mixtures we used took $0.15\%$ the compute of the training run to find and improved the negative log likelihood by $1-5\%$. For our chemistry experiments we considered mixing assay datasets in PubChem (Beaini et al., 2023; Kim et al., 2016) to improve the performance of an XGBoost (Chen & Guestrin, 2016) model on several held-out assay datasets. In all cases, we observed MixMin improved over the natural distribution (the standard data mixture after filtering for this domain (Salem et al., 2020; Li et al., 2022; Ye et al., 2018)) as we increased the number of surrogates to mix over, improving average precision scores between $0.03 - 0.15$. We note an additional benefit of optimizing data mixtures for chemistry tasks is that the found MixMin weights could provide interpretability; we highlight patterns MixMin found in PCBA for predicting

assays (e.g., distinct but predictive cytotoxicity assays). To summarize, our contributions are:

1. Observing data mixing reduces to a single convex minimization as model classes become more expressive

2. Proposing MixMin which approximately solves the convex objective by using cheap proxy models and downstream data samples

3. Empirically showing MixMin improves past baselines for finding data mixtures across language modeling and chemistry tasks

4. Empirical and analytical evidence showing MixMin is robust to the use of weak/cheap proxy models

## 2. Preliminaries

We denote a hypothesis space by $\mathcal{H}$, a model by $f : \mathcal{X} \to \mathcal{O} \in \mathcal{H}$, a datapoint by $(x, y) \in \mathcal{X} \times \mathcal{Y}$, and our loss function by $\mathcal{L}(o, y) : \mathcal{O} \times \mathcal{Y} \to \mathbb{R}^+$. This paper considers the problem of mixing a set of finite source distributions ($dp \in P$ where $P$ is finite) to train on to do better on some downstream "target" distribution ($dt$). For example, mixing Wikipedia and arXiv data to do better on benchmarks evaluating scientific knowledge.

Specifically, we are searching for the best weighting $\lambda \in \Delta^P$ (where $\Delta^P$ is the simplex) of our sources to train a model on to perform well on our downstream distribution. Formally, using our loss function $\mathcal{L}$ as both our measure of downstream performance and training objective:

$$DM(\lambda, \mathcal{H}) = \int_{\mathcal{X} \times \mathcal{Y}} \mathcal{L}(f_\lambda(x), y) dt(x, y)$$

where $f_\lambda(x) = \arg\min_{f \in \mathcal{H}} \sum \lambda_p \int_{\mathcal{X} \times \mathcal{Y}} \mathcal{L}(f(x), y) dp(x, y)$

(Data Mixing)

In general $\min_\lambda DM(\lambda, \mathcal{H})$ a bi-level optimization that can be hard to optimize: a standard approach is grid/random searching through mixture weights and solving the inner

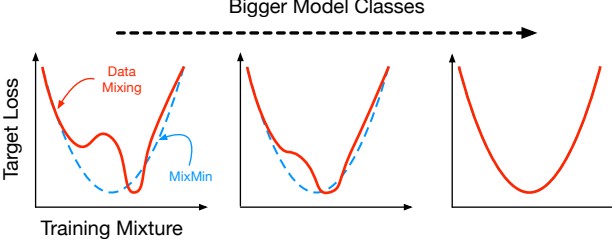

Figure 2: The convex MixMin objective better approximates the Data Mixing objective as the model class becomes larger (and better approximates Bayes optimal).

minimization by fitting a (small) proxy model. In this paper we describe settings where the bi-level optimization collapses and we can use simple gradient based optimization to find the best mixture.

**Data Filtering is not Data Mixing** An often related but different approach to curating training data is data filtering. Abstractly, data filtering is some map $F$ that takes a dataset $D \to D'$ where $D' \subset D$. Such approaches are motivated by training efficiency, e.g., coresets, and/or data quality, e.g., rejection sampling to be closer to a desired target distribution. Data filtering can be composed with data mixing; one typically mixes amongst filtered data sources (Dubey et al., 2024). We leave studying such compositions to future work, but note some popular examples of data filtering include: perplexity correlations (Thrush et al., 2024), n-gram or embedding similarity with a target distribution (Xie et al., 2023; Everaert & Potts, 2024), handcrafted classifiers (Xiao et al., 2024; Li et al., 2024a), and perplexity filtering (Li et al., 2024a)

## 3. Data Mixing is Learning a Linear Model

We show that Data Mixing simplifies to a convex optimization problem as the model classes become larger. This intuition is presented in Figure 2.

### 3.1. Well-Specified: Data Mixing is just Risk Minimization

We show that, when our hypothesis space contains the Bayes optimal model $f_p$ for all $dp \in P$, the best Data Mixing weights are also the best weighting of models trained on each source (a convex objective). The core idea is that for certain losses, cross-entropy (CE) and mean-squared error (MSE), the Bayes optimal model for a mixture is the mixture of the Bayes optimal models' for the sources. However this is only true if there is no input distribution shift among the sources ($p(x)$ are all the same), e.g., for generative tasks. To clarify, MixMin holds for the unconditional CE loss, $\int -log(f(x))dp(x)$, which matches $f(x)$

to $dp(x)$. MixMin also holds for conditional CE and MSE losses under no covariate shift: $\int -log(f_y(x))dp(x, y)$ and $\int ||f(x) - y||_2^2 dp(x, y)$, with $dp(x) = dp'(x) \; \forall dp, dp' \in P$.

**Theorem 3.1.** *Let the objective for Data Mixing be unconditional CE, $\int -log(f(x))dp(x)$, or conditional CE or MSE with no covariate shift, $\int -log(f^y(x))dp(x, y)$ and $\int ||f(x) - y||_2^2 dp(x, y)$, with $p(x) = p'(x) \; \forall p, p' \in P$. Suppose also $\mathcal{H}$ contains the Bayes optimal model for each mixture of the source distributions $dp \in P$. Then $\lambda^* = \arg\min_{\lambda \in \Delta^P} DM(\lambda, \mathcal{H})$ iff*

$$\lambda^* = \arg \min_{\lambda \in \Delta^P} \int_{\mathcal{X} \times \mathcal{Y}} \mathcal{L} \left( \sum \lambda_p f_p(x), y \right) dt(x, y) \quad (1)$$

The proof is provided in Appendix A.1. This optimization is clearly convex, and can be handled with gradient based approaches unlike Data Mixing. However, running this optimization requires having the Bayes optimal $f_p$ and an estimator for the integral to compute the gradients. The main hurdle will be how to get reasonable approximations for the Bayes optimal $f_p$, which we discuss in Section 4.

### 3.2. Not Well-Specified: 'Expressivity' is Enough

We now show that Data Mixing smoothly approaches Equation 1 as the hypothesis space $\mathcal{H}$ contains models closer Bayes optimal, e.g., becomes larger. This follows from assuming some regularity in our loss function and the definition of the data mixing objective.

**Lemma 3.2.** *For $\mathcal{L}$ either CE or MSE, suppose $\int \mathcal{L}(f(x), y)dt(x, y)$ is $C$-Lipschitz in $f$ w.r.t a norm on the functions $||f(x)||$ [1]. Let*

$$\lambda^* = \arg \min_{\lambda \in \Delta^P} \int_{\mathcal{X} \times \mathcal{Y}} \mathcal{L} \left( \sum \lambda_p f_p(x), y \right) dt(x, y).$$

*Then, for any hypothesis space $\mathcal{H}$ such that for all mixtures $dp_\lambda \in Conv(P)$, the $dp_\lambda$ risk minimizer $\hat{f}_\lambda(\mathcal{H})$ in $\mathcal{H}$ satisfies $||\hat{f}_\lambda(\mathcal{H}) - f_\lambda|| \leq \epsilon$ where $f_\lambda$ is the Bayes optimal model [2], we have the excess Data Mixing error is bounded by $\epsilon$:*

$$DM_{\mathcal{H}}(\lambda^*, dt) - \min_{\lambda \in \Delta^P} DM_{\mathcal{H}}(\lambda, dt) \leq 2C\epsilon \quad (2)$$

The proof is provided in Appendix A.2. The main intuition to take away from Lemma 3.2 is that as our model classes

---

[1] To ensure this for MSE one could consider only bounded output spaces, and for cross-entropy, bounded away from 0 and 1.

[2] Effectively this means functions too far away will not have low error. Mathematically, this means there is lower-bound on the increase in loss from Bayes optimal in terms of the norm of the perturbation to the function (e.g., strong-convexity).

---

**Algorithm 1** MixMin

---

**Require:** Step size $\eta$, number of steps $n$, loss function $\mathcal{L}$ (either cross-entropy or $\ell_2^2$), samples $D_t$ from the target distribution $dt$, and (cheap) models trained on each source $\hat{f}_p(x) \ \forall \ dp \in P$.

**Initialize:** $\lambda_p \leftarrow \frac{1}{|P|}$ for all $dp \in P$, and pre-compute $\{\hat{f}_p(x) \ \forall \ x \in D_t, dp \in P\}$

1: **for** $i = 1, \ldots, n$ **do**
2:     $\hat{f}_\lambda(x) \leftarrow \sum_{dp \in P} \lambda_p \hat{f}_p(x)$
3:     $l \leftarrow \frac{\lambda_p}{|D_t|} \sum_{(x,y) \in D_t} \mathcal{L}(\hat{f}_\lambda(x), y)$
4:     $g \leftarrow \nabla_\lambda l$
5:     $\lambda_p \leftarrow \frac{\lambda_p e^{\eta g_p}}{\sum_{dp \in P} \lambda_p e^{\eta g_p}}$ for all $dp \in P$
6: **end for**
   Return$\{\lambda_p\}_{dp \in P}$

---

get larger and becomes closer to expressing the Bayes optimal model, the Equation 1 weights become closer to the optimal Data Mixing weights. This is depicted in Figure 2

## 4. MixMin

We now consider how to optimize the objective in Equation 1. If we had a gradient oracle, we could optimize the objective using entropic descent (Duchi, 2018). However, the objective (and gradient) is typically intractable for several reasons which we now alleviate with approximations.

Firstly, computing the integral for the objective (and gradient) is typically intractable. We can however compute an empirical risk given a dataset $D_t \sim dp_t$. Secondly, we often do not know the Bayes optimal models $f_p$ for each $dp \in P$, but instead can obtain "approximations" $\hat{f}_p$, e.g., by training a model on a dataset $D_p \sim dp \in P$. This gives us our approximate convex mixture objective, where we let Train$(\mathcal{H}, D_p)$ be some training function:

$$
\begin{aligned}
&\lambda^*(\{\hat{f}_p\}_{p \in P}) \\
&= \arg\min_{\lambda \in \Delta^P} \frac{1}{|D_t|} \sum_{(x,y) \in D_t} \mathcal{L}(\sum_{p \in P} \lambda_p \hat{f}_p(x), y) \\
&\qquad \text{where } \hat{f}_p = \text{Train}(\mathcal{H}, D_p)
\end{aligned}
$$

$$\text{(MixMin)}$$

Computing the gradient of the objective follows from chain rule, and hence we can run entropic descent to solve MixMin. Recall entropic descent is a variant of mirror descent which handles the constraint of being in the simplex. This gives us Algorithm 1, which we call MixMin for 'Mixtures by Minimization'.

**Cost** We only need to evaluate $f_p(x) \ \forall x \in D_t$ once and can reuse the outputs for subsequent calls to the gradient. Given this, the dominating cost with MixMin, as with grid/random search based approaches to data mixing, is the number of models needed to train (i.e., $\hat{f}_p$). MixMin requires only as many models as sources, while the number of models grid/random search approaches require scales exponentially (in the worst-case) with the number of sources. Note once the outputs and proxy models are computed, the per-iteration cost of MixMin is $O(|P||D_t|)$, which is independent of model size.

**On the use of weak proxy models** In our experiments, we found that proxy models $\hat{f}_p$ trained with very small fractions of total training compute achieved good results and additional proxy model compute did not significantly improve the quality of the final mixtures. This suggests that MixMin can perform well when the proxy models $\hat{f}_p$ have a high loss and that the excess error of MixMin should be analyzed beyond the risk of the proxy models $\hat{f}_p$. Specifically, we empirically found that we could train with significantly less data (1%) and still have similar MixMin performance. Furthermore, we found the ensemble of the proxy models could be far less accurate than the model retrained on the MixMin mixture weights, suggesting the proxy models were far from Bayes optimal [3] but still led to good mixtures. We hope future work investigates and generalizes these phenomenon.

## 5. Related Work

Some past work has considered optimizing a pretraining data mixture for a specific downstream loss, i.e., task. Hwang & Whang (2021) train many small models with different mixtures and learn to predict the task error of unseen mixtures, and RegMix (Liu et al., 2024) later used a similar oracle to find better mixtures (the best mixture if the oracle is accurate). Similarly, Held et al. (2025) proposed optimizing the mixture given estimates of downstream performance from different source datasets; they assumed data mixing was a linear model (with constraints) in the mixture weights and proposed mixtures by maximizing the performance of the linear model. We note Liu et al. (2024) considered modeling data mixing as a linear objective, but found their non-linear models for predicting data mixing improved over linear models. Finally there are the normal grid-search approaches (Blakeney et al., 2024), which random search is known to improve (Bergstra & Bengio, 2012).

Another line of work has considered data mixing objectives defined by just the sources and not a downstream loss: e.g., with the motivation of faster optimization or generalizing

---

[3] Being close to Bayes optimal would imply the ensemble was close to retraining, a contradiction.

better across the sources. However, when evaluating on a downstream objective (as is the focus of our paper) methods here are known to not consistently improve over the natural or balanced distribution of data (Fan et al., 2023; Jiang et al., 2024; Albalak et al., 2023; Xie et al., 2024): see Held et al. (2025) for comparisons. Outside of language modeling, we note the natural distribution of data is the standard data mixture used for transfer learning in chemistry (after filtering data) (Salem et al., 2020; Li et al., 2022; Ye et al., 2018). Another line of work considered data mixing for distributionally robust optimization (Thudi & Maddison, 2024), which is distinct to downstream data mixing objective.

Given these past findings, we compare to RegMix and random search in our experiments. However, compared to MixMin, RegMix has significantly higher evaluation costs.

Our setting for Bi-Level optimization also presents several challenges for previous approaches, including those used for other hyperparameter optimizations. Firstly, the inner-optimization is over an arbitrary function space (e.g., non-parametric models like XGBoost), and so lacks the parametric gradients needed for many methods (Ji et al., 2021; Pedregosa, 2016). Alternative constraint based approaches through KKT suffer similar issues given the non-parametric space of the inner optimization (Shi et al., 2005). For many model classes, one could parameterize the inner optimization, but we then lose the necessary convexity for these methods (e.g., Neural Networks).

## 6. Experiments

We compared MixMin to previous data mixing baselines for language modeling and chemistry tasks. For language modeling we further investigated the transferability of mixtures found using cheap proxy models to more expensive training runs. For chemistry, we explored the performance of MixMin as we increased the pool of surrogate data sources. Experiments were run using A100 GPUs and AMD EPYC 7643 CPUs.

Our experiments support the following list of (weak) conditions for when MixMin works:

1. The data sources do not have covariate shift (consider changing the objective to remove covariate shifts)

2. The downstream task is close to/contained within a mixture of our sources

3. Our final model class is very expressive

4. Our proxy model's learnt some signal for each source (e.g., even with very little data): consider iteratively using more compute for the proxy models until stability in the mixtures is reached

### 6.1. Pretraining Language Models

Data mixing is now a common step in pre-training LLMs (Dubey et al., 2024; Li et al., 2024b). We investigated how MixMin compared to other data mixing methods, its compute efficiency, and how weights found at a smaller scale of models transfers to training larger models [4].

#### 6.1.1. EXPERIMENTAL SETUP

**Datasets and Models** We used the domains in SlimPajama (cer, 2023) as sources for pre-training. For the downstream target, we considered several multiple-choice tasks that models in the $160M - 410M$ scale are known to do non-trivially at (Thrush et al., 2024), alongside another generative task: SciQ (Welbl et al., 2017), PIQA (Bisk et al., 2020), ARC-Easy (Clark et al., 2018) and the first 10000 documents in OpenWebMath (Paster et al., 2023) [5]. For multiple-choice tasks we used MixMin to maximize the log probability of the correct question and answer (Q+A) sequence, i.e., treating these tasks as new pretraining domains. Note that the optimal conditional of A given Q distribution is a conditional of the joint distribution of Q+A, so our objective subsumes the $Q \rightarrow A$ prediction problem. Alongside the objective loss, we also evaluated accuracy and the conditional cross-entropy of the pre-trained models for these tasks (i.e., multiple-choice performance). OpenWebMath is a specialized pretraining dataset, and so the loss for MixMin and evaluation was the negative log probability of a sequence. Our experiments used the $160M$ and $410M$ pythia models (Biderman et al., 2023) trained for $3.2B$ and $8.2B$ tokens respectively (chinchilla optimal (Hoffmann et al., 2022)).

**Data Filtering** We included documents in SlimPajama with at least 1024 tokens (using the Pythia tokenizer), and scraped documents for each domain until adding the next document would bring us over 3.2B tokens for each domain.

**Hyperparameter Tuning** We took the largest batch size that fits on a single A100 gpu, which was 64 for the $160M$-pythia model and 32 for the $410M$-pythia model for a context length of 1024. For the $160M$-pythia model we increased the learning rate until training loss got worse on the natural distribution of domains in SlimPajama: we started from $1e-4, 5e-4, 1e-3, 5e-3, 1e-2$ and found loss got worse at $1e-2$ so chose $5e-3$. For the $410M$-pythia model we evaluated learning rates $5e-3$ and $1e-2$ and found $5e-3$ was better. Other hyperparameters are the same as (Thrush et al., 2024). All hyperparemeters are fixed throughout the language modeling experiments.

---

[4] $p$ values from Welch's t-test are shown in Appendix B.
[5] We chose 10000 documents for computational efficiency.

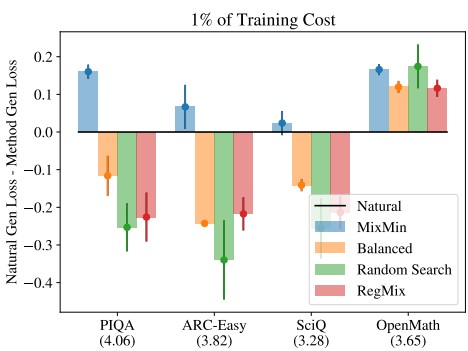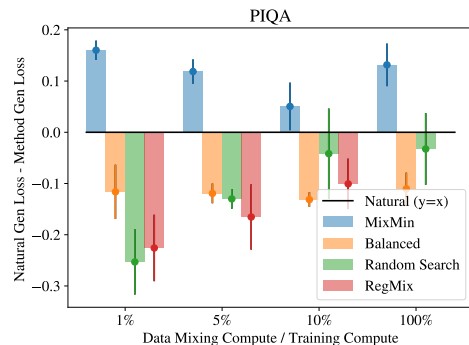

Figure 3: **MixMin consistently outperforms all baselines across the four target tasks using** $1\%$ **of the final training run compute (Left).** We report improvement over the downstream generative loss of training on the natural distribution (which is stated beside the task name): higher is better. Error bars indicate a $95\%$ confidence interval. **Furthermore, we find that MixMin was robust to using less compute (Right), while RegMix and random search had their performance degrade with less compute.**

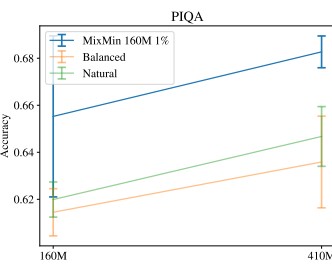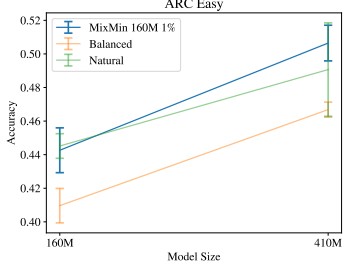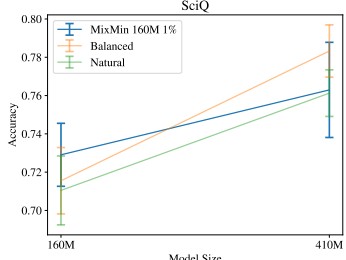

Figure 4: **MixMin mixtures derived from small models continue to improve training for larger models as measured by target accuracy (in most cases).** We report accuracy with the errors bars representing a $95\%$ confidence interval over 3 trials. MixMin weights were found using $1\%$ the compute of the $160M$ model training run, which is $\sim 0.15\%$ the compute of the $410M$ training run.

**Method Implementations** We compared MixMin to the baselines of balancing the domains, the natural distribution of domains in SlimPajama, random search and RegMix (the latter known to improve or perform on par as previous data mixing methods). We split the target task into a random $80\%$ training set, and $20\%$ test set. The training set was used for the MixMin optimization, and the evaluation of loss for random search and RegMix. Results are reported over 3 trials of random train-test splits.

We ran MixMin using $\eta = 1.0$ for 100 steps for all the experiments. For RegMix, we adapted the code available at `https://github.com/sail-sg/regmix/blob/main/regression_fitting/regression.ipynb`, changing the hard-coded natural distribution to the SlimPajama natural distribution and inputting our models' results. We implemented random search by sampling uniformly from the simplex and training proxy models on those mixture. Recall MixMin requires a proxy model for each domain, and to normalize evaluation

compute we ran random search with the same number of proxy models (7). RegMix showed benefits by using many (but cheap) proxy models (Liu et al., 2024), and so we ablated the number of our cheapest proxy models to vary compute for it; this however increased its evaluation costs to the other baselines. All proxy models are trained with the same hyperparameters specified earlier, and use the $160M$ pythia model architecture. To achieve $X\%$ compute relative to a full training run of the $160M$ pythia model, we trained each proxy model for $3.2(X/100)(1/7)B$ tokens for MixMin and random search, and for RegMix train $7X$ proxy models where each proxy model trains for $3.2(1/700)B$ tokens (our lowest compute proxy models).

### 6.1.2. RESULTS

**MixMin found good mixtures with** $1\%$ **of train compute** In Figure 3 we show the generative loss results of MixMin, random search, and RegMix where the total compute for all the proxy models was $1\%$ relative to training the final

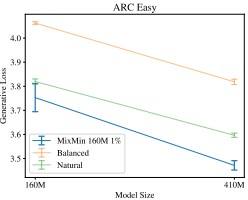 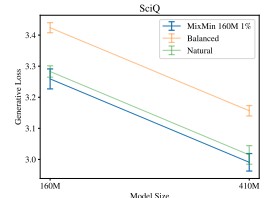 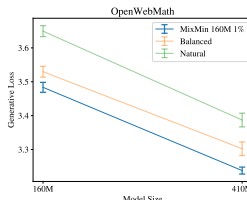

Figure 5: **MixMin mixtures derived from small models continue to improve training for larger models as measured by target generative loss (in all cases).** We report generative loss with the errors bars representing a $95\%$ confidence interval over 3 trials (*lower is better*). MixMin weights were found using $1\%$ the compute of the $160M$ model training run, which is $0.15\%$ the compute of the $410M$ training run.

$160M$-pythia model for $3.2B$ tokens (Chinchilla optimal). We compared the performance of these to a balanced distribution and the natural distribution. We found MixMin consistently improved generative loss over the baselines. We also found MixMin consistently matched or improved the predictive loss and accuracy over all the baseline methods (Figure 8 in Appendix B).

**MixMin was more cost effective than baselines** In figure 3 (and Figure 9 in Appendix B for other tasks) we compared the performance of MixMin, random search, and RegMix as we vary the the total compute for all the proxy models from $100\%, 10\%, 5\%, 1\%$ of the compute for training the final $160M$-pythia model for $3.2B$ tokens (Chinchilla optimal). We found MixMin was robust to using less compute, where as random search and RegMix benefitted from increasing compute (but do not meet MixMin performance even at the highest compute setting we tested). Note, we did not include RegMix at $100\%$ compute given the significant evaluation overhead: 100 times the alternatives.

**MixMin mixtures for smaller models improve training of larger models** Finally, we tested whether MixMin mixtures derived from smaller models were useful for training larger models. We computed MixMin mixtures using pythia-$160M$ proxy models each trained with $0.14\%$ the compute of a full $3.2B$ token training run (for an overall cost of $1\%$). We then trained a pythia-$410M$ for $8.2B$ tokens on those mixtures. Here we only compared to the balanced or natural mixture, as our previous experiments showed random search and RegMix found mixture comparable or worse than one of those mixtures for our tasks. Accuracy results are presented in Figure 4 with generative and predictive loss results presented in Figure 5 and Figure 12 (the latter in Appendix B). For accuracy and predictive loss we found in most cases MixMin's performance improved over the baselines as we increased scale, despite not refitting the MixMin weights. Even in the cases where MixMin's accuracy and predictive loss did not strictly improve over baselines, *the generative loss always improved with MixMin.*

## 6.2. Chemistry

We explored how MixMin could improve predicting properties of molecule libraries, in particular the endpoints of an assay (a predictive task). Many assays have few positive samples and training on just that data can lead to only noisy performance. Past work has remedied this by transfer learning from a larger dataset, but had not considering optimizing the data mixture after filtering (Salem et al., 2020; Li et al., 2022; Sun et al., 2022; Ye et al., 2018). A common default mixture is the natural distribution, and here we show MixMin provided better mixtures than this, especially as the diversity of the source datasets grows. We found similar improvements over random search and RegMix. We hope MixMin's ability to scale helps future data curation efforts for predictive drug screening.

### 6.2.1. EXPERIMENTAL SETUP

**Datasets and Models** We worked with the PCBA dataset (Beaini et al., 2023) [6], which was a filtered scrape of PubChem (Kim et al., 2016). We used the first 10 bioassays as the target distributions, using subsets of all the bioassays (removing the target) as sources. Our experiments used XGBoost on ECFP fingerprints of SMILES (Rogers & Hahn, 2010) to fit each assay, known to be a strong baseline (Yang et al., 2019; Seidl et al., 2023) for this domain. We report AP performance (average precision score, a variant of AUPR) for our experiments.

**Data Filtering** We trained models over the first 100,000 molecules in PCBA. We further removed any assay with no positive or negative instances among the first 100,000 molecules. For all assays (1328) the number after filtering was 1281, for the first 100 assays this left 86, and the first 10 were all kept. We performed data mixing among the filtered assays (excluding the target assay).

---

[6]The dataset can be found at https://polarishub.io/datasets/graphium/pcba-1328-1564k-v1.

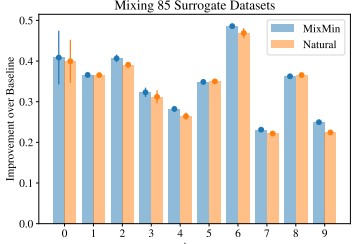
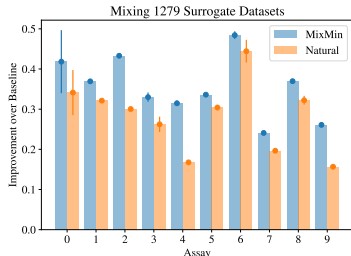

Figure 6: **MixMin improved over using the natural distribution of data as we increased the number of surrogate assays.** We report AP scores for the first 10 assays in PCBA, with error bars representing a $95\%$ confidence interval over 3 trials.

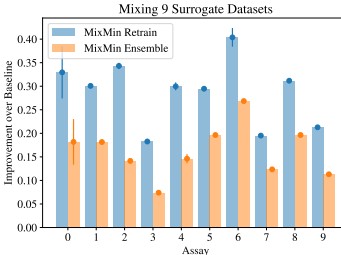
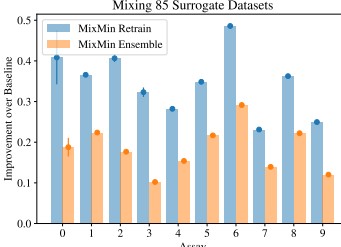
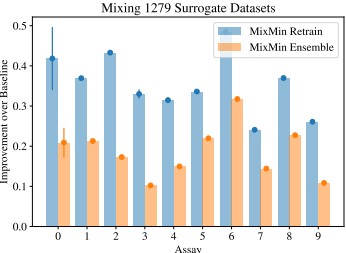

Figure 7: **Retraining with the MixMin mixture performs better than using the MixMin weights to ensemble the proxy models.** We report AP scores with error bars representing a $95\%$ confidence interval over 3 trials. *This implies the proxy models were far from Bayes optimal.*

**Method Implementation** For every assay in PCBA, we used a $64\% - 16\% - 20\%$ train-validation-test split: an original $80\% - 20\%$ train-test split, and further splitting the train set into a $20\%$ validation set. We trained proxy models used for MixMin on just the train split, and trained baseline models on the train and validation splits. The baselines models represent the performance we get by just training on the target, as to normalize improvement from data mixing. Given a target assay, we used the proxy models on the other assays in PCBA to run MixMin, using the train set on the target task to fit the MixMin mixture weights (leaving the validation and test set unseen). Finally, given the MixMin mixture we retrained a new model on the train sets from all the other assays. For the natural distribution baseline, we trained on all the train sets from other assays as is (no reweighting of sources). For all method we reported the mean test AP score improvement (from 3 trials) over the baseline of training on the target distribution, with $95\%$ confidence intervals.

**Hyperparameter Tuning** We selected hyperparameters for each proxy and baseline model by doing a 5-fold cross-validation on the train set. We grid search over all combinations of $n\ estimators \in [10, 50, 100]$ and $max\ depth \in [4, 6, 8]$. For the models trained over all the surrogate assays

(the final model trained on MixMin or natural mixtures) we fixed the $n\ estimators = 100$ and $max\ depth = 6$.

### 6.2.2. RESULTS

**MixMin's advantage over the baselines increased with the number of sources** As shown in Figure 6, we observed that MixMin performs mostly on par with the natural distribution when working with the first 10 assays and improved over natural as the number of assays to mix grew. This suggests MixMin was able to find relevant data sources amongst more diverse sets, and we note its absolute performance grew while the performance of the natural mixture decreased with the number of sources. We note in all cases both MixMin and training on the natural distribution improved over the baseline of training on the target assay. Similar improvements over random search and RegMix are presented in Figure 16 and Figure 17 respectively (in Appendix B).

**Retraining with MixMin Mixture performed better than ensembling proxy models** An alternative approach to training on MixMin weights, suggested by the reduction for Bayes optimal models, is to ensemble the proxy models with the MixMin weights. We present a comparison between this and retraining with MixMin weights in Figure 7. Retraining

performed significantly better, suggesting the proxy models were too noisy to lead to good ensembles. Specifically, this implies the proxy models were far from Bayes optimal, as the ensemble would have then also been near Bayes optimal. The fact we were able to improve over the natural distribution suggests MixMin did not need strong proxy models for these sources and targets.

**MixMin highlighted data sources with similar endpoint measure** All the target assays tested cytotoxicity, and the majority of the top 4 data sources MixMin found for each assay also tested cytotoxicity (Table 2 in Appendix B). More specific protein inhibition assays were also included, potentially providing insights into the compound's mechanism of action. However, an assay with no apparent connection to the target assay was also identified. We hope future work explores the full interpretability of MixMin weights for the bioassays we tested on.

## 7. Conclusion

In this paper we formalized a bi-level data mixing objective for mixing data sources to minimize risk on a downstream objective. We showed that in the case of CE or MSE, this objective reduces to a convex minimization as our model classes become larger. We proposed using cheap proxy models trained on the individual source distributions to optimize the convex objective, leading to a data mixing method we call MixMin. Our experiments showed MixMin consistently outperformed previous baselines across language modeling and chemistry tasks, and was robust to using cheap proxy models. We hope future work explores explaining when and why MixMin can use cheap proxy model, or develops alternative empirical approaches to optimizing the convex objective for data mixing with large model classes.

**Limitations** Our reductions for data mixing were specific to CE and MSE, and required there be no covariate shift among the source distribution. While this covers generative tasks, we hope future work explores extending these ideas to other loss functions, and handles covariate shift. For example, our framework does not currently allow us to do data mixing for image classification, where covariate shift is common among sources. Also, MixMin need not be the best way to optimize the convex data mixing objective. Finally, an open problem is pushing the limit of how cheap the proxy model's can be without degradation to performance.

## Impact Statement

This paper presents work whose goal is to advance the field of Machine Learning. There are many potential societal consequences of our work, none which we feel must be specifically highlighted here.

## Acknowledgements

Resources used in preparing this research were provided in part by the Province of Ontario, the Government of Canada through CIFAR, and companies sponsoring the Vector Institute. We acknowledge the support of the Natural Sciences and Engineering Research Council of Canada (NSERC), RGPIN-2021-03445. Anvith Thudi is supported by a Vanier Fellowship from NSERC.

Evianne Rovers acknowledges support given through the Matthieu Schapira lab at the University of Toronto who receives funding from NSERC [Grant RGPIN-2019-04416], CQDM (Quantum Leap-176), and MITACS accelerate (IT13051). The Structural Genomics Consortium is a registered charity (no: 1097737) that receives funds from Bayer AG, Boehringer Ingelheim, Bristol Myers Squibb, Genentech, Genome Canada through Ontario Genomics Institute [OGI-196], EU/EFPIA/OICR/McGill/KTH/Diamond Innovative Medicines Initiative 2 Joint Undertaking [EUbOPEN grant 875510], Janssen, Merck KGaA (aka EMD in Canada and US), Pfizer, and Takeda.

We thank Chris Crebolder and the rest of the UofT Computer Science sysadmin team for their help in making many of the experiments in this paper possible. We would also like to thank Marta Skreta, Leo Cotta, Ayoub El Hanchi, William Held, and many others at the Vector Institute for discussions contributing to this paper.

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

# A. Proofs

## A.1. Theorem 3.1

*Proof.* Recall that the Bayes optimal model for conditional cross entropy is $p(y|x)$ and for conditional MSE is $\mathbb{E}_{y \sim p(y|x)} y$. In both cases, letting $f_\lambda$ be the Bayes optimal for the mixture $dp_\lambda = \sum_{p \in P} \lambda_p dp$, and $f_p$ be the Bayes optimal for the individual sources, we have for cross-entropy and $\ell_2^2$:

$$f_\lambda = \frac{\sum_{dp \in P} \lambda_p f_p(x) p(x)}{\sum_{dp' \in P} \lambda_{p'} p'(x)}$$

and in particular, when there is no covariate shift amongst the sources we have $f_\lambda = \sum_{dp \in P} \lambda_p f_p(x)$.

Note that the Bayes optimal for unconditional CE is just p(x), and so similarly $f_\lambda(x) = \sum_{dp \in P} \lambda_p f_p(x)$.

With this formula for $f_\lambda$, we then have our Data Mixing objective reduces to just learning a linear model over $f_p(x)$. Specifically plugging in $f_\lambda = \sum_{dp \in P} \lambda_p f_p(x)$ into Data Mixing gives Equation 1.

$\square$

## A.2. Lemma 3.2

*Proof.* First note that, using the Lipschitz criterion with the condition on the hypothesis space and the definition of data mixing, letting $\mathcal{H}^*$ contain the Bayes optimal for all mixtures, we have $\forall \lambda \in \Delta^P$ that

$$|DM_\mathcal{H}(\lambda, dt) - DM_{\mathcal{H}^*}(\lambda, dt)| \leq C\epsilon.$$

Specifically, let $f_{H,\lambda}$ and $f_{H^*,\lambda}$ be the minimizer of the mixture $\lambda$ in $H$ and $H^*$ respectively. By definition of $H^*$, $f_{H^*,\lambda}$ is Bayes optimal, and by the assumption in the lemma statement, $||f_{H,\lambda} - f_{H^*,\lambda}|| \leq \epsilon$. Now note $|DM_H(\lambda, dt) - DM_{H^*}(\lambda, dt)| = |\int \mathcal{L}(f_{H,\lambda}(x), y) dt(x, y) - \int \mathcal{L}(f_{H^*,\lambda}(x), y) dt(x, y)| \leq C ||f_{H,\lambda} - f_{H^*,\lambda}||$ by the Lipschitz assumption. Combining the two inequalities we have $|DM_H(\lambda, dt) - DM_{H^*}(\lambda, dt)| \leq C\epsilon$ which gives the desired inequality.

This relation will be enough for the proof, and hence weaker criteria in the lemma that ensure this relation will suffice.

Let $\bar{\lambda} = \arg\min_{\lambda \in \Delta^P} DM_\mathcal{H}(\lambda, dt)$ be the optimal data mixture weights and recall $\lambda^* = \arg\min_{\lambda \in \Delta^P} DM_{\mathcal{H}^*}(\lambda, dt)$ where $\mathcal{H}^*$ contains the Bayes optimal functions for all mixtures. Now note the left hand side of the inequality

$$DM_\mathcal{H}(\lambda^*, dt) - \min_{\lambda \in \Delta^P} DM_\mathcal{H}(\lambda, dt) = DM_\mathcal{H}(\lambda^*, dt) - DM_{\mathcal{H}^*}(\lambda^*, dt) + DM_{\mathcal{H}^*}(\lambda^*, dt) - DM_\mathcal{H}(\bar{\lambda}, dt).$$

The first difference is bounded by $C\epsilon_1$ by the earlier inequality for changing hypothesis spaces. Now bounding the second difference we have

$$DM_{\mathcal{H}^*}(\lambda^*, dt) - DM_\mathcal{H}(\bar{\lambda}, dt) = DM_{\mathcal{H}^*}(\lambda^*, dt) - DM_{\mathcal{H}^*}(\bar{\lambda}, dt) + DM_{\mathcal{H}^*}(\bar{\lambda}, dt) - DM_\mathcal{H}(\bar{\lambda}, dt)$$
$$\leq DM_{\mathcal{H}^*}(\bar{\lambda}, dt) - DM_\mathcal{H}(\bar{\lambda}, dt) \leq C\epsilon_1$$

where the first inequality came from the definition of $\lambda^*$, and the second from the inequality stated at the beginning of the proof for changing hypothesis spaces.

Hence we conclude

$$DM_\mathcal{H}(\lambda^*, dt) - \min_{\lambda \in \Delta^P} DM_\mathcal{H}(\lambda, dt) \leq 2C\epsilon_1$$

$\square$

# B. Extra Tables and Figures

| Target | First | Second | Third | Fourth |
|--------|-------|--------|-------|--------|
| assayID-1 | assayID-891 | assayID-620 | assayID-618 | assayID-693 |
| assayID-3 | assayID-620 | assayID-5 | assayID-618 | assayID-891 |
| assayID-5 | assayID-620 | assayID-891 | assayID-92 | assayID-425 |
| assayID-7 | assayID-620 | assayID-952 | assayID-693 | assayID-618 |
| assayID-9 | assayID-891 | assayID-693 | assayID-620 | assayID-256 |
| assayID-11 | assayID-620 | assayID-952 | assayID-693 | assayID-618 |
| assayID-13 | assayID-618 | assayID-710 | assayID-891 | assayID-451 |
| assayID-15 | assayID-19 | assayID-620 | assayID-25 | assayID-758 |
| assayID-17 | assayID-19 | assayID-618 | assayID-758 | assayID-21 |
| assayID-19 | assayID-25 | assayID-17 | assayID-952 | assayID-21 |

Table 1: The top 4 assays found by MixMin (over the 1328 assays in PCBA) for each target task. We list the PubChem assayID corresponding to the assay in PCBA. Red indicated assays measuring cytotoxicity, green are phenotypic screens, blue are specific protein binding/inhibition assays, and orange are other types of assays.

| Target | First | Second | Third | Fourth |
|--------|-------|--------|-------|--------|
| assayID-1 | 0.036705 | 0.036184 | 0.035904 | 0.030910 |
| assayID-3 | 0.05026778 | 0.04660426 | 0.0414932 | 0.03289703 |
| assayID-5 | 0.05026778 | 0.04660426 | 0.0414932 | 0.03289703 |
| assayID-7 | 0.03170111 | 0.02936895 | 0.02839234 | 0.02707635 |
| assayID-9 | 0.03645385 | 0.02978397 | 0.02941164 | 0.02894659 |
| assayID-11 | 0.04995361 | 0.03905968 | 0.03367052 | 0.03212925 |
| assayID-13 | 0.05764348 | 0.04289928 | 0.03077811 | 0.02817639 |
| assayID-15 | 0.11996419 | 0.04302281 | 0.03967432 | 0.03457846 |
| assayID-17 | 0.22989839 | 0.04496526 | 0.04188494 | 0.03894247 |
| assayID-19 | 0.10650713 | 0.05704838 | 0.05019803 | 0.04435973 |

Table 2: The mixing weights for the top 4 assays found by MixMin (over the 1328 assays in PCBA) for each target task. We list the PubChem assayID corresponding to the target assay in PCBA.

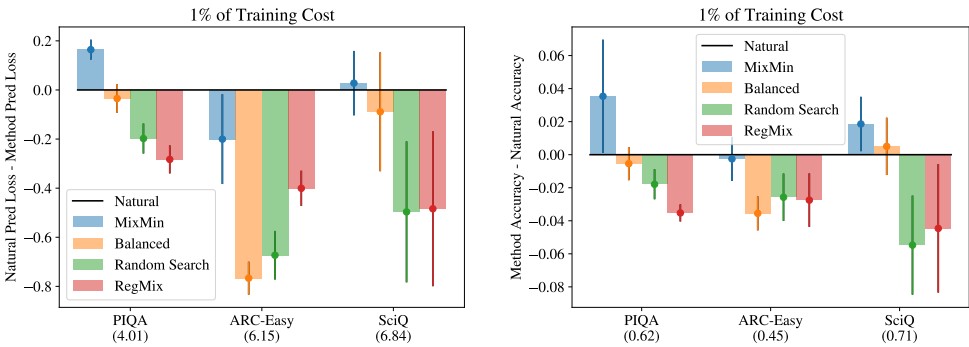

Figure 8: In almost all cases MixMin improves or matches all baselines across the target tasks using 1% of the final training run compute (Left). We report improvement over the predictive loss (left) and accuracy (right) of training on the natural distribution, which is stated on the x-axis with the task name.

| Baseline | PIQA | ARC-EASY | SciQ | Open WebMath |
|---|---|---|---|---|
| Random Search | 0.006 | 0.011 | 0.019 | 0.839 |
| RegMix | 0.007 | 0.004 | 0.003 | 0.051 |
| Balanced | 0.008 | 0.013 | 0.006 | 0.027 |
| Natural | 0.025 | 0.206 | 0.375 | 0.0003 |

Table 3: $p$-values for a Welch's t-test with the null hypothesis being MixMin performs the same as the baselines for the 1%-compute experiments presented in Figure 3 (generative loss) over the 3 trials used.

| Baseline | 1% | 5% | 10% | 100% |
|---|---|---|---|---|
| Random Search | 0.006 | 0.0003 | 0.233 | 0.043 |
| RegMix | 0.007 | 0.011 | 0.023 | NA |
| Balanced | 0.008 | 0.0003 | 0.017 | 0.002 |
| Natural | 0.025 | 0.004 | 0.224 | 0.018 |

Table 4: $p$-values for a Welch's t-test with the null hypothesis being MixMin performs the same as the baselines on PIQA for generative loss across compute budgets (as shown in Figure 3) over the 3 trials used.

| Baseline | PIQA | ARC-EASY | SciQ | Open WebMath |
|---|---|---|---|---|
| Balanced | 0.019 | 0.000 | 0.003 | 0.020 |
| Natural | 0.052 | 0.003 | 0.409 | 0.002 |

Table 5: $p$-values for a Welch's t-test with the null hypothesis being MixMin performs the same as the baselines on generative loss across tasks for the $410M$ models experiments shown in Figure 5.

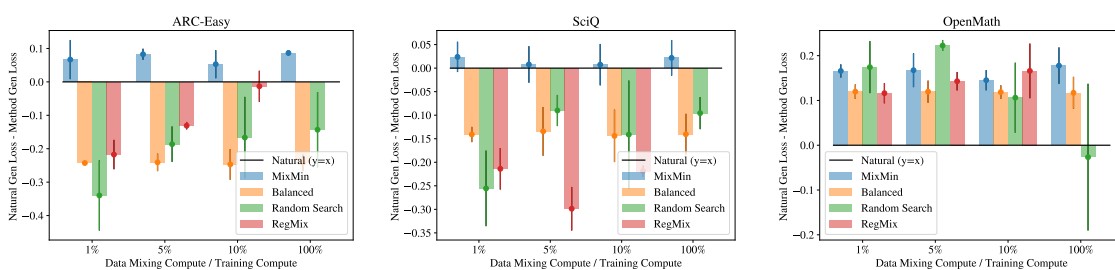

Figure 9: MixMin consistently performed better than alternative methods on generative loss across compute budgets, and was robust to using less compute. We note both Random Search and RegMix tend to improve with more compute. We report improvement over the generative loss of training on the natural distribution.

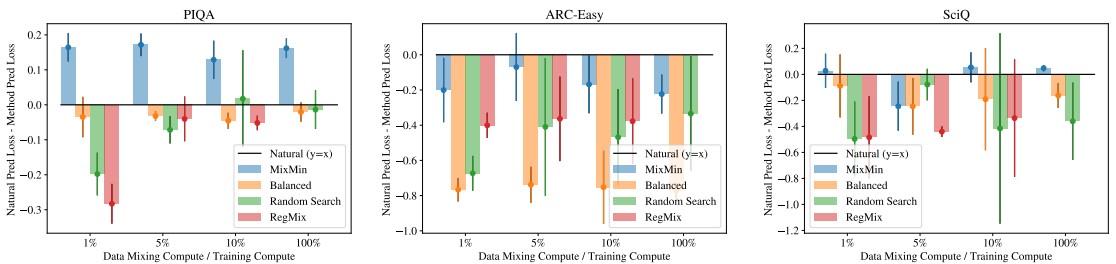

Figure 10: MixMin often performed better or on par to alternative methods on predictive loss across compute budgets, and was robust to using less compute. We note both Random Search and RegMix tend to improve with more compute. We report improvement over the predictive loss of training on the natural distribution.

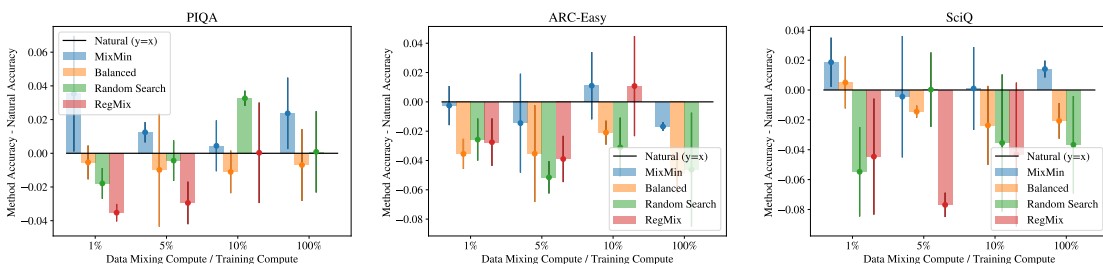

Figure 11: MixMin often performed better or on par to alternative methods on accuracy across compute budgets, and was robust to using less compute. We note both Random Search and RegMix tend to improve with more compute. We report improvement over the accuracy of training on the natural distribution.

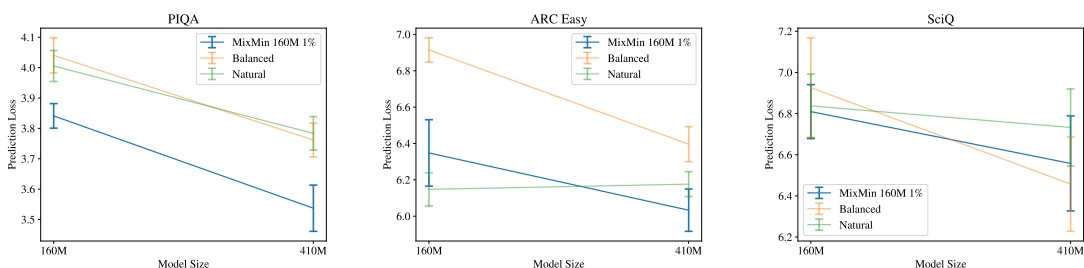

Figure 12: In most cases MixMin consistently improved predictive loss over the baselines as we scaled the models. We report predictive loss relative to a pythia-$160M$ model trained with the natural distribution of tokens (lower in the figures is better). MixMin weights were found using $1\%$ the compute of the $160M$ model training run, which is $0.15\%$ the compute of the $410M$ training run.

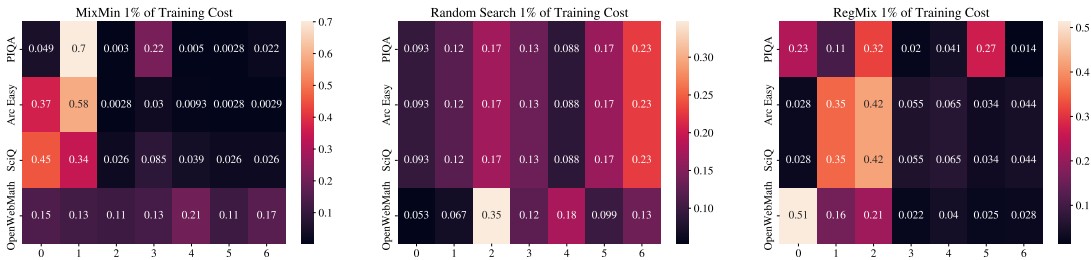

Figure 13: The mixture weights found by MixMin, Random Search, and RegMix when using $1\%$ the compute for a pythia-$160M$ training run on $3.2B$ tokens. Note the domains (x-axis) are "RedPajamaCommonCrawl", "RedPajamaC4", "RedPajamaGithub", "RedPajamaBook", "RedPajamaArXiv", "RedPajamaWikipedia", "RedPajamaStackExchange" respectively.

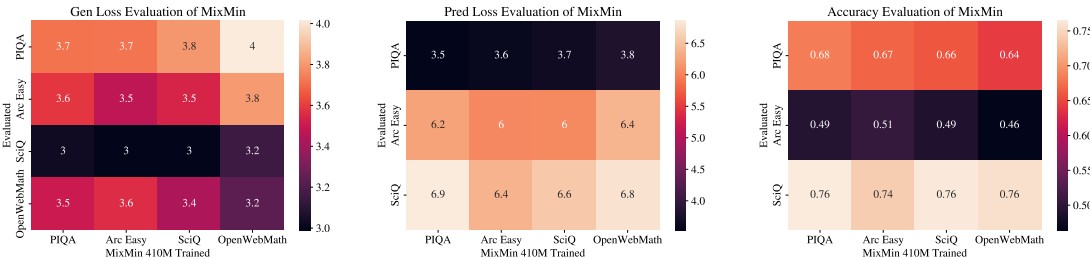

Figure 14: We found the MixMin models for PIQA, Arc Easy, and SciQ perform similarly across tasks, however were (almost) always best for their own task across metrics. We report the cross performance of MixMin for different tasks at the 410M parameter scale. The subfigures report generative loss, predictive loss, and accuracy respectively.

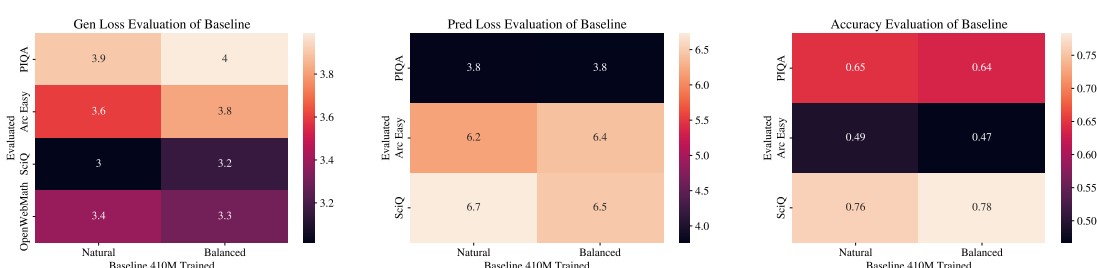

Figure 15: We report the performance of training on the natural and balanced mixture for different tasks at the 410M parameter scale. The subfigures report generative loss, predictive loss, and accuracy respectively. Comparing to Figure 14 we see both the natural and baseline mixture are matched or dominated in average performance across tasks by the MixMin weights for PIQA, Arc Easy, and SciQ.

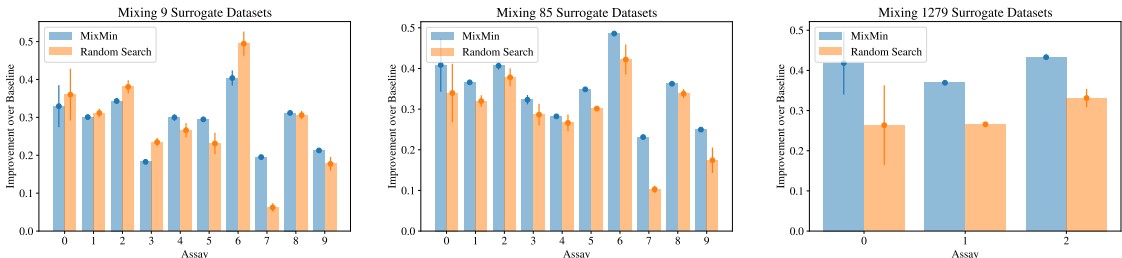

Figure 16: **MixMin improved over using random search as we increased the number of surrogate assays.** We report AP scores for the first 10 assays in PCBA, with error bars representing a 95% confidence interval over 3 trials. We only report the first 3 assays for the 1279 sources setting.

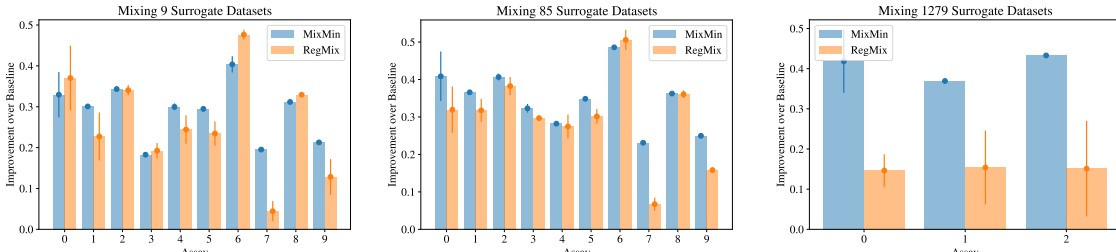

Figure 17: **MixMin improved over using RegMix as we increased the number of surrogate assays.** We report AP scores for the first 10 assays in PCBA, with error bars representing a $95\%$ confidence interval over 3 trials. We only report the first 3 assays for the 1279 sources setting.

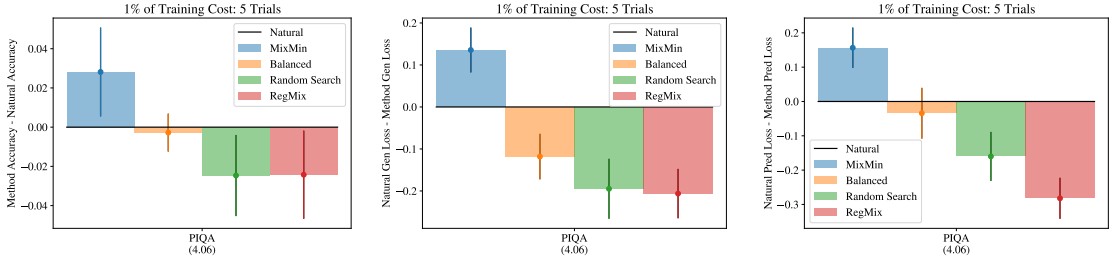

Figure 18: MixMin often performed better or on par to alternative methods on accuracy, generative loss, and predictive loss on PIQA for the $160M$ model using $1\%$ compute for proxy models. Here we present results over 5 trials.

