# OpenReview forum: "MixMin: Finding Data Mixtures via Convex Minimization"
_ICML.cc/2025/Conference — ICML 2025 poster_

### Official Review · Reviewer_nsAa · 2025-02-26

**Overall Recommendation:** 3

**Summary:**

This submission addresses the optimization of data source mixtures, formulating it as a bi-level optimization problem. The key result is Theorem 3.1, which states that under certain conditions (cross-entropy or mean squared error loss, hypothesis class contains Bayes optimal models), the optimal mixture weights are the same as those for optimally mixing the outputs of Bayes optimal models for each data source. The authors then propose the MixMin method as a practical realization of the key result, where less expensive proxy models are substituted for Bayes optimal models. Experiments are conducted on language model pre-training for better downstream task performance and on transfer learning for chemical assay prediction. MixMin's data mixtures achieve larger gains than the baselines considered in these experiments.

### Update after rebuttal

Thanks to the authors for their follow-up reply. The verbatim changes to the theorems and proofs are good and I approve them.

I am still uneasy about reporting confidence intervals or performing statistical tests on only 3 experiment trials. I recommend that the authors take the time before the camera-ready submission to increase the number of trials to at least 5, for all experiments, and perform Welch's t-tests as in the rebuttal and/or recompute confidence intervals.

Overall, I am placing not insignificant trust in the authors to carry out the revisions (including the clarifications on experiments and bi-level optimization mentioned previously) and hope that they will do so for the camera-ready version.

**Claims And Evidence:**

On the theoretical side, I have issues with the proof of Lemma 3.2 and the applicability of the results to generative models. On the experimental side, I think that 3 train-test splits is too few and have several questions about the experimental design. Please see the respective sections below.

**Essential References Not Discussed:**

Gradient-based methods for bi-level optimization (please see the previous comment).

**Experimental Designs Or Analyses:**

I reviewed all of the experiments section (Section 6) and think the following issues are important:
1. I think 3 train-test splits is too few. It does not seem meaningful to show 95% confidence intervals from three splits. How are these computed?
1. For the language model multiple-choice tasks, why choose the loss to be the (negative) log probability of the question and correct answer (Q + A) sequence, as opposed to the probability of A given Q? It seems that evaluating the model on also generating the question Q makes the task harder than necessary.
1. I do not understand the claim that the RegMix baseline has higher computational cost. As I understand it, for RegMix, the total number of tokens for training proxy models is (3.2B / 700 tokens/model) * (7X models) = 3.2B * (X / 100), which is the same as the (3.2B X / 700 tokens/model)  * (7 models) for MixMin and random search.
1. For the chemical assay datasets, I do not understand the choices of training data that were made.
    1. What is meant by "baseline models"? Why are they trained on the train + validation splits, while MixMin's proxy models are trained only on the train splits? What prevents MixMin's proxy models from using train + validation also?
    1. Given the MixMin mixture, does the training data for the final model include the train split (or train + validation) for the target assay as well? I have the same question as above (train only vs. train + validation) for all assays included in this final training.
1. For the chemical assay experiment, why is the natural distribution the only baseline considered? Why not include the same baselines from before: balanced distribution, random search, RegMix?

**Methods And Evaluation Criteria:**

The proposed MixMin method makes sense for data mixture optimization thanks to the connection given by Theorem 3.1. The two experimental evaluations, data source optimization for language model pre-training and transfer learning for chemical prediction, are also sensible.

**Other Comments Or Suggestions:**

Minor comments and questions:
- Line 84, right column: Should "a set of finite source distributions" be "a finite set of source distributions," i.e., "finite" applies to the set, not the distributions?
- Lines 85-86, right column: Related to the previous comment, it is unclear at this point what the notation $dp$, $dt$ means.
- Data Filtering is not Data Mixing: Is data filtering (at the data source level) not a special case of data mixing where some mixture weights are set to zero and the remaining non-zero weights are equal? Is the distinction being drawn here more about data sample level vs. source level?
- Lines 171-172, right column: What is meant by "the excess error of MixMin should be analyzed beyond the risk of the proxy models $\hat{f}_p$?
- Lines 325-326, left column: Does "predictive loss" mean cross-entropy with respect to the answer choices? This could be clarified.
- Lines 335-336, right column: Does "trained models over the first 100,000 molecules in PCBA" mean the first 100,000 molecules in each assay?
- I would appreciate a little more information about the chemical assay datasets. What is the data type of the target variable? Is the data tabular or otherwise structured?
-
- Figures 6 and 7: I suggest labelling the x-axis as "target array" for clarity.
- Lines 432-433, left column: Should Table 2 be Table 1?

**Other Strengths And Weaknesses:**

I think the key result of Theorem 3.1 is also the main strength of the work, showing that under some conditions, it may be possible to avoid the bi-level optimization inherent to data mixture optimization and instead optimize the mixture of outputs of source-specific models.

**Questions For Authors:**

Please see:
- Points 2-5 under Experimental Designs or Analyses
- Major issues 1, 2 under Theoretical Claims (although the latter may require more involved responses)

Also from "Relation to Broader Scientific Literature":
- To what extent are gradient-based methods for bi-level optimization applicable?

**Relation To Broader Scientific Literature:**

I question the claim that bi-level optimization "generally only admits expensive zero-order approaches" (lines 19-21). A quick search yields survey/tutorial papers such as [Zhang et al. (2023)](https://arxiv.org/abs/2308.00788) that discuss gradient-based methods, based on implicit gradients or unrolling of the lower-level optimization. I think it is a significant omission to not at least discuss such methods. Furthermore, while gradient-based methods may make assumptions that do not hold for the data mixing problem here, it may still be possible to run the algorithms and compare results (particularly on the chemical assay datasets that I presume involve less computation than language modeling). Doing so would better contextualize MixMin's performance in terms of mixture quality and/or computational cost.

**Theoretical Claims:**

I checked the proofs of Theorem 3.1 and Lemma 3.2. I have two major issues and three minor ones.

Major issues:
1. For the proof of Lemma 3.2, it is not clear how the first displayed inequality is obtained, which is key for the remainder of the proof. More steps should be shown, and perhaps the authors have to be more precise about which norm is used for functions $f$.
1. Although this is not a formal theoretical result, I do not understand the claim in lines 157-159 that "distributions for generative tasks are only over 'labels' $p(y)$." This statement should be explained as it supports the application of MixMin to language modeling.

Minor issues:
1. "No input distribution shift/covariate shift" should be formally defined in Section 3.1 as it is not clear without reading the proof of Theorem 3.1.
1. Footnote 2: I do not see how the claim that "functions too far away will not have low error" follows from the near-Bayes optimality assumed in Lemma 3.2.
1. Lines 572-573: What does "left hand sign of the inequality" mean?

---

> ### Author Rebuttal · Authors · 2025-03-31
>
> We thank the reviewer for their time and feedback. We discuss the main questions below, and will incorporate the other suggestions to our revised draft.
>
> > For the proof of Lemma 3.2, it is not clear how the first displayed inequality is obtained…
>
> We describe the derivation below, and will add this to our revised draft.
>
> Let $f_H$ and $f_{H^*}$ be the minimizer of the mixture $w$ in $H$ and $H^*$ respectively. By definition of $H*$, $f_{H^*}$ is Bayes optimal, and by the assumption in the lemma statement, $||f_H - f_{H^*}|| \leq \epsilon$.  Now note  $|DM_{H}(w)- DM_{H^*}(w)| = | Loss(f_H) - Loss(f_{H^*})| \leq C ||f_H - f_{H^*}||$ by the Lipschitz assumption. Combining the two inequalities we have $|DM_{H}(w) - DM_{H^*}(w)| \leq C \epsilon$, which gives the desired inequality.
>
> > I do not understand the claim in lines 157-159…
>
> We mean MixMin holds for the unconditional CE loss, $\int -log(f(x)) dp(x)$, which matches $f(x)$ to $dp(x)$. Mixmin also holds for conditional CE and MSE losses under no covariate shift: $\int -log(f_{y}(x)) dp(x,y)$ and $\int ||f(x) -y||_2^2 dp(x,y)$, with $dp_i(x) = dp_j(x)$ for all sources. We will add these precise statements to our revised draft.
>
> > I think 3 train-test splits is too few…
>
> We evaluated MixMin over many downstream tasks, and sources (with 3 trials for each setting). Over these hundreds of experiments we saw MixMin consistently improved over the baselines.  As a sanity check, we also increased the number of trials for our PIQA experiments with 160M models to 5 trials, and found the results were consistent. See the figures here: https://anonymous.4open.science/r/MixMin_LLM_extra_trials-54C6
>
> > Why choose the loss to be the (negative) log probability of the question and correct answer…
>
> The A|Q task is a conditional CE task that can have covariate shifts (different probability of Qs among sources): MixMin does not apply. Hence we chose to learn A+Q which is an unconditional CE task and so MixMin applies. See our response to Reviewer KxeH for more discussion on the issue of non-convexity for covariate shift data mixing.
>
> > I do not understand the claim that the RegMix baseline has higher computational cost…
>
> The difference is the evaluation cost. RegMix requires evaluating 7X models (where X can be 10 in our experiments) but MixMin requires evaluating just 7.
>
> > What is meant by "baseline models"? Why are they trained on the train + validation splits, while MixMin's proxy models are trained only on the train splits?..
>
> We meant models (XGBoost) trained solely on the downstream task data, using the train + validation splits for that task. MixMin models are trained on the source datasets
>
> We left the validation sets untouched to allow for unforeseen future experiments that require a validation set (e.g., additional hyperparameter tuning on the MixMin mixture). In this paper the default XGBoost hyperparameters on the MixMin mixture already worked better than the baseline model (with hyperparameters selected from cross validation), so they were unused.
>
> > Does the training data for the final model include the train split (or train + validation) for the target assay as well?
>
> No, it does not include the target assay. The train split of the target was only used to tune the MixMin weights.
>
> > For the chemical assay experiment, why is the natural distribution the only baseline considered?...
>
> Our language modeling experiments already showed the other baselines often performed worse than natural. We have now tested RegMix and Random Search on the chemistry experiments. We observed the same trends as natural; the baselines become worse while MixMin becomes better with increasing sources (only tested 3 targets for 1280 sources due to compute). The figures are here: https://anonymous.4open.science/r/MixMin_Chem_More_Baselines-286C/
>
> > To what extent are gradient-based methods for bi-level optimization applicable?
>
> The methods from this literature were not applicable to our problem, so we did not discuss them. To clarify this we plan to include the following paragraph in our revised draft:
>
> “Our setting for Bi-Level optimization presents several challenges for previous approaches, including those used for other hyperparameter optimizations. Firstly, the inner-optimization is over an arbitrary function space (e.g., non-parametric models like XGBoost), and so lacks the parametric gradients needed for many methods [1,2]. Alternative constraint based approaches through KKT suffer similar issues given the non-parametric space of the inner optimization [3]. For many model classes, one could parameterize the inner optimization, but we then lose the necessary convexity for these methods (e.g., Neural Networks).”
>
> [1] K. Ji et al “Bilevel optimization: Convergence analysis and enhanced design”
>
> [2] F. Pedregosa, “Hyperparameter optimization with approximate gradient”
>
> [3] C. Shi et al “An extended kuhn–tucker approach for linear bilevel programming”

---

> > ### Comment · Reviewer_nsAa · 2025-04-03
> >
> > Thanks to the authors for their responses. Below are my follow-up comments:
> >
> > 1. **Proof of Lemma 3.2:** I think I understand now the justification for the first inequality. However, my review of this was hindered by the different notations for $DM$: $DM(\lambda, \mathcal{H})$ in the (Data Mixing) equation, $DM_{\mathcal{H}}(\lambda, dt)$ in Lemma 3.2, $DM_{\mathcal{H}}(w)$ in the rebuttal. The notation should be made consistent, in addition to writing down the additional steps in the derivation.
> > 1. **Claim in lines 157-159:** I think now that there are three cases: MSE, "conditional" CE involving two variables $x, y$, and "unconditional" CE involving only $x$ or only $y$. These should be distinguished and precisely defined (with equations). Then I would like to see the statement and proof of Theorem 3.1 be refined to reflect these three cases, where the "no covariate shift" assumption should also be clarified and is needed only for the MSE and conditional CE cases. This will also better justify the use of the joint Q + A loss for the language model experiment rather than the conditional A | Q.
> > 1. **Train-test splits and confidence intervals:** I do not understand why the confidence intervals in e.g. Figure 3 in the paper, with 3 trials, appear no larger and in some cases clearly smaller than the confidence intervals in the corresponding rebuttal figure with 5 trials (for example, the blue MixMin bar in Figure 3). Thus, I still question whether confidence intervals obtained from only 3 trials are meaningful.
> >
> > The rebuttal addresses my other questions about the experiments and gradient-based methods for bi-level optimization, provided that the clarifications are added to the paper.
> >
> > Overall, I am maintaining my score, to be interpreted as "minor revision", because I would personally like to see a revised manuscript that addresses remaining issues 1-3 above before recommending acceptance.

---

> > > ### Author Response · Authors · 2025-04-04
> > >
> > > Thank you for your response! We provide below the verbatim updated text as requested, though are not allowed by ICML policy to share a revised manuscript.
> > >
> > > > I think I understand now the justification for the first inequality. However, my review of this was hindered by the different notations for DM…
> > >
> > > Sorry for the notation change, we were character limited in our original response. Below is the derivation with consistent notation, which will be included in the proof.
> > >
> > > Let $f_{H,\lambda}$ and $f_{H^*,\lambda}$ be the minimizer of the mixture $\lambda$ in $H$ and $H^*$ respectively. By definition of $H^*$, $f_{H^*,\lambda}$ is Bayes optimal, and by the assumption in the lemma statement, $||f_{H,\lambda} - f_{H^*,\lambda}|| \leq \epsilon$.  Now note $|DM_{H}(\lambda,dt) - DM_{H^*}(\lambda,dt)| = | \int \mathcal{L}(f_{H,\lambda}(x),y) dt(x,y)- \int \mathcal{L}(f_{H^*,\lambda}(x),y) dt(x,y)| \leq C ||f_{H,\lambda} - f_{H^*,\lambda}||$ by the Lipschitz assumption. Combining the two inequalities we have $|DM_{H}(\lambda,dt) - DM_{H^*}(\lambda,dt)| \leq C \epsilon$ which gives the desired inequality.
> > >
> > > > I think now that there are three cases … Then I would like to see the statement and proof of Theorem 3.1 be refined to reflect these three cases, where the "no covariate shift" assumption should also be clarified and is needed only for the MSE and conditional CE cases.
> > >
> > > Below are the refined theorem statements and proofs. Note the proof is mostly the same except for an additional line clarifying the unconditional CE case.
> > >
> > > **Theorem 3.1**: Let the loss function for Data Mixing be unconditional CE, $\int_{\mathcal{X} \times \mathcal{Y}} \mathcal{L}\left(f(x),y\right) dp(x,y) = \int_{\mathcal{X}} -log(f(x)) dp(x)$, or conditional CE or MSE with no covariate shift, $\int_{\mathcal{X} \times \mathcal{Y}} -log(f^{y}(x)) dp(x,y)$ or $\int_{\mathcal{X} \times \mathcal{Y}} ||f(x) -y||_2^2 dp(x,y)$ with $dp(x) = dp'(x)~\forall p,p' \in P$. Suppose also $\mathcal{H}$ contains the Bayes optimal model for each mixture of the source distributions $dp \in P$.
> > >
> > > Then $\lambda^* = \arg \min_{\lambda \in \Delta^P} DM(\lambda, \mathcal{H})$ iff $\lambda^* =  \arg\min_{\lambda \in \Delta^{P}} \int_{\mathcal{X} \times \mathcal{Y}} \mathcal{L}\left(\sum \lambda_p f_p(x),y\right) dt(x,y)$.
> > >
> > > **Proof**: Recall that the Bayes optimal model for conditional cross entropy is $p(y|x)$ and for conditional MSE is $\mathbb{E}_{y \sim p(y|x)} y$.
> > >
> > > In both cases, letting $f_{\lambda}$ be the Bayes optimal for the mixture $dp_{\lambda} = \sum_{p\in P} \lambda_{p} dp$, and $f_p$ be the Bayes optimal for the individual sources, we have for cross-entropy and $\ell_2^2$: $f_{\lambda} = \frac{\sum_{dp \in P} \lambda_p f_p(x) p(x)}{\sum_{dp' \in P} \lambda_{p'} p'(x)}$  and in particular, when there is no covariate shift amongst the sources we have $f_{\lambda} = \sum_{dp \in P} \lambda_p f_p(x)$.
> > >
> > > Note that the Bayes optimal for unconditional CE is just $p(x)$, and so similarly $f_{\lambda}(x) =  \sum_{dp \in P} \lambda_p f_p(x)$.
> > >
> > > With this formula for $f_{\lambda}$, we then have our Data Mixing objective reduces to just learning a linear model over $f_p(x)$. Specifically, plugging in $f_{\lambda} = \sum_{dp \in P} \lambda_p f_p(x)$ into Data Mixing gives Equation 1.
> > >
> > > >  I do not understand why the confidence intervals in e.g. Figure 3 in the paper, with 3 trials, appear no larger and in some cases clearly smaller than the confidence intervals in the corresponding rebuttal figure with 5 trials (for example, the blue MixMin bar in Figure 3). Thus, I still question whether confidence intervals obtained from only 3 trials are meaningful.
> > >
> > > We have now conducted a Welch’s t-test between MixMin and the baselines for our 3 trial setup (on the generative loss results for Pythia-160M on PIQA using 1% to train proxy models) and found the MixMin improvements were statistically significant. All cases had a p value under $0.05$.
> > >
> > > 1) MixMin vs. Random Search: $p = 0.006$
> > > 2) MixMin vs. RegMix: $p = 0.007$
> > > 3) MixMin vs. Balanced: $p = 0.008$
> > > 4) MixMin vs. Natural: $p = 0.025$
> > >
> > > In the 5 trial setup, we see the p values are also reduced as expected (unlike the confidence intervals which seemed mostly the same). The 5 trial results are:
> > >
> > > 1) MixMin vs. Random Search: $p =0.0003$
> > > 2) MixMin vs. RegMix: $p =0.00007$
> > > 3) MixMin vs. Balanced: $0.0004$
> > > 4) MixMin vs. Natural: $p = 0.0142$
> > >
> > > Note these 5 trial hypothesis tests reuse data from the 3 trial test, so we need to be cautious when making claims with both. We will include tables with the p values for all our experiments (using 3 trials) in the Appendix in our revised draft.

---

### Official Review · Reviewer_YnxL · 2025-03-15

**Overall Recommendation:** 3

**Summary:**

This paper proposes MixMin, a simple but effective method for solving the data mixture coefficients in large language model pretraining. The authors identify that the bi-level optimization objective for solving data mixture is intractable. But luckily, such an objective tends to be convex when model classes become larger. Thus, such an objective can be effectively solved using a gradient-based approach. Some theoretical justifications are provided along with experimental evaluations for LLM pretraining experiments to justify the effectiveness of the proposed MixMin method.

**Claims And Evidence:**

The claim of MixMin is that it improves model performance (in terms of better benchmark evaluation scores) with marginal additional computing overhead. This claim is well supported by the empirical evaluations.

**Essential References Not Discussed:**

N/A

**Experimental Designs Or Analyses:**

The experimental design and analyses make sense and are pretty thorough.

**Methods And Evaluation Criteria:**

Both the proposed method and evaluation criteria make sense and follow the common practice of the LLM-related research field.

**Other Comments Or Suggestions:**

Please see the ones in "Strengths And Weaknesses".

**Other Strengths And Weaknesses:**

Strengths:
- The paper is well-written and well-motivated.
- Optimizing data mixture for LLM pretraining is an impactful research direction.
- The proposed method is intuitive and easy to understand.
- Both theoretical and empirical evaluations have been provided to justify the performance of the proposed method.

Weaknesses:
- Only models under the parameter size of 1B have been evaluated. It's not clear if the method scales well to larger model sizes, such as around 7B.
- It seems that the experiments directly use domains contained in SlimPajama. I wonder if the method also works for data domains like web crawl data, code, books, arXiv/Wikipedia, etc., which are more common in LLM pretraining practices.
- It seems that the major computational overhead comes from training small LLMs on sub-data domains. I wonder how large the small-scale LLMs need to be and what can be done when there are many data domains.

**Questions For Authors:**

Please see the ones in "Strengths And Weaknesses".

**Relation To Broader Scientific Literature:**

The proposed method can potentially improve the data curation and training recipe of large language models or foundation models in general.

**Theoretical Claims:**

I checked the proofs of Theorem 3.1 and Lemma 3.2; both seem to make sense.

---

> ### Author Rebuttal · Authors · 2025-03-31
>
> We thank the reviewer for their time and feedback. We discuss specific questions below.
>
> > Only models under the parameter size of 1B have been evaluated. It's not clear if the method scales well to larger model sizes, such as around 7B.
>
> We agree it would be nice to extend our analysis to larger scale experiments, but at the moment, we do not have the computational resources to run those experiments. We did find MixMin’s relative performance to be stable over the two scales that we tested (160M, 410M), and also in non LLM domains (chemistry). Although it's far from conclusive, it suggests that the benefits of MixMin may continue to hold.
>
> >It seems that the experiments directly use domains contained in SlimPajama. I wonder if the method also works for data domains like web crawl data, code, books, arXiv/Wikipedia, etc., which are more common in LLM pretraining practices.
>
> SlimPajama domains already contain the mentioned categories. The seven domains in SlimPajama are: CommonCrawl, C4, Github, Books, arXiv, Wikepedia, and Stackexchange. We also tested MixMin on sources from a completely different domain (chemistry), with varying combinations of sets, and MixMin continued to perform well.
>
> > It seems that the major computational overhead comes from training small LLMs on sub-data domains. I wonder how large the small-scale LLMs need to be and what can be done when there are many data domains.
>
> On the question of many domains, in our chemistry experiments we tested MixMin across many sets of sources, ranging from 9 to 1280 sources. We found MixMin improved as the number of sources grew.
>
> In terms of complexity, we found MixMin was more efficient than past approaches. MixMin still needs to train a proxy model for each source which scales with the number of sources, but this is true of past approaches. Importantly, we found the proxy models for MixMin could be obtained cheaply by using smaller models and less data (see our LLM experiments where we used <1% of the cost of the full training run to compute all the proxy models).
>
> Further experiments could explore pushing the cost of data mixing smaller than what we tested (<1% of the full training run), perhaps to find when MixMin would break. We will acknowledge this open problem in a revised draft.

---

### Official Review · Reviewer_KxeH · 2025-03-24

**Overall Recommendation:** 4

**Summary:**

The authors propose a novel method (called MixMin) for the problem of optimizing data mixtures for pre-training of large ML models in order to improve performance in downstream tasks. The MixMin method proposed by the authors solves this optimization problem with the following approach: First, MixMin trains a cheap/small “proxy model” on each of the individual data sources. Then, MixMin learns the best mixture of the “proxy model” outputs for the target dataset. This problem of finding the best mixture of “proxy model” outputs is a convex optimization problem, so MixMin solves this problem using simple entropic descent. Finally, MixMin outputs the optimized mixture of “proxy model” outputs as the optimal mixture of the corresponding data sources used to train each “proxy model”. The resulting mixture of data sources found by this MixMin method can then be used to train the larger, complex ML model that we want to pre-train and apply to the downstream task. The authors show that the larger the ML model of interest, the better the convex optimization problem solved by the MixMin method approximates the true data mixing objective. Thus, for models that are sufficiently large, solving the true data mixing optimization problem effectively becomes a simple convex optimization problem using the MixMin method. The authors demonstrate empirically that using MixMin to find optimal data mixtures improves performance of models on downstream tasks in language modeling and chemistry. Empirical results show that MixMin consistently outperforms baseline approaches for optimizing data mixtures, and that MixMin can find good data mixtures with significantly smaller computational cost compared to other approaches.

## update after rebuttal
As I stated in my comment below, the authors answered all of my questions in their rebuttal, and I agree with the author's proposal of adding the provided list of "weak conditions for when MixMin works" to the paper. I therefore maintain that this work should be accepted for all of the reasons stated in my initial review.

**Claims And Evidence:**

Yes.

**Essential References Not Discussed:**

There are no essential references missing as far as I am aware.

**Experimental Designs Or Analyses:**

Yes, I checked all experimental design setups and experimental results provided in the paper. All are valid as far as I am aware.

**Methods And Evaluation Criteria:**

Yes.

**Other Comments Or Suggestions:**

Typos:

1: Line 125/126: "filering" should be "filtering"

2: Line 209/210: "resuse" should be "reuse"

3: Line 572/573: "left hand sign" should probably be "left hand side"?

**Other Strengths And Weaknesses:**

Strengths:

Strength 1: Relevancy of problem setting: The problem of finding an optimal mixture of data to pre-train models efficiently is clearly a relevant problem that can improve performance of models in downstream tasks across a wide variety of applications and ML pipelines.

Strength 2: Empirical results: The empirical results in the paper provide convincing evidence that the proposed MixMin method consistently outperforms other methods for finding optimal data mixtures across a variety of downstream tasks. This is further strengthened by the inclusion of performance on relevant, real-world chemistry tasks, and by the additional results demonstrating that MixMin can achieve good mixtures with smaller computation cost.

Strength 3: Quality of Writing: The writing is very clear and easy to understand.

Weaknesses:

A minor weakness of this paper is that the novel methodological contribution is on the smaller side because the proposed MixMin method is a fairly simple approach to optimizing data mixtures by training proxy models and then optimizing a mixture with entropic descent. That being said, this is a very minor weakness because the authors show that the MixMin method works very well empirically, consistently outperforming other approaches and cutting computation cost. I therefore still recommend that this paper be accepted.

**Questions For Authors:**

Question 1:
Can the authors provide any additional insight into when they expect MixMin to work well in practice? In what scenarios (i.e. what types of data sources, downstream tasks, etc.) do the authors expect that using these cheap proxy models and applying the MixMin method will be successful? In what scenarios do the authors expect that MixMin would not be useful (i.e. data sources with covariate shift)? It would be nice to have a concrete list of criteria for problems such that the authors think that MixMin would be useful. Such a list would allow practitioners to more easily determine if MixMin is applicable to their particular problem setting, so I think adding something like this would strengthen the paper.

Question 2:
Do the authors have any concrete ideas of how MixMin could be adapted in the future to handle data sources with covariate shift (i.e. image classification)? To be clear, I don’t think that concrete ideas for this are necessary to include in this paper, but I am intellectually curious.

**Relation To Broader Scientific Literature:**

Finding an optimal data mixture for machine learning pipelines is a very relevant problem to the broader ML research community because improving the data mixture used to train a model can significantly improve performance for downstream tasks. This problem is a particularly relevant problem for researchers who work on training and applying LLMs. The author’s proposed MixMin method for solving this problem is therefore relevant to the broader research community, especially since the authors provide convincing empirical evidence that MixMin consistently outperforms baseline approaches.

**Theoretical Claims:**

Yes, I checked the proofs in Appendix Section A and they are correct as far as I am aware.

---

> ### Author Rebuttal · Authors · 2025-03-31
>
> We thank the reviewer for their time and feedback! We elaborate on questions raised in the review below, and will update the draft with the typos pointed out by the reviewer.
>
> > Can the authors provide any additional insight into when they expect MixMin to work well in practice? In what scenarios (i.e. what types of data sources, downstream tasks, etc.) do the authors expect that using these cheap proxy models and applying the MixMin method will be successful? In what scenarios do the authors expect that MixMin would not be useful (i.e. data sources with covariate shift)?
>
> If our proxy models are sufficiently close to Bayes optimal, then MixMin will find the near optimal mixture weights. This is if there is no covariate shift among our sources, and that we ultimately train another near Bayes optimal model on the mixture (i.e., big data and big model regime). Note, if the downstream task is not close to any mixture, then MixMin (or any data mixing) may not help performance on the downstream task.
>
> The main question is when are the proxy models “practically” sufficiently close to Bayes optimal for the MixMin optimization. Our language modeling experiments showed that using 1% for the proxy models is comparable to using 100% the compute of a full training run. We believe this suggests a general approach to selecting the compute for the proxy models–we could iteratively increase the proxy model compute and run MixMin and see when the mixture becomes stable across scale. Ultimately, we did not need to do this as the cheapest settings we tested already performed better than the baselines.
>
> > It would be nice to have a concrete list of criteria for problems such that the authors think that MixMin would be useful.
>
> We thank the reviewer for the suggestion of providing a list of criteria for using MixMin. Given our previous discussion we propose to add the  following list to the paper:
>
> “(Weak) Conditions for when MixMin works:
>
> 1) The data sources do not have covariate shift (or we can change the learning problem to not have covariate shift)
>
> 2) We believe the downstream task is close to/contained within a mixture of our sources
>
> 3) Our final model class is very expressive (i.e., do not have significant architectural biases)
>
> 4) Our proxy model learnt some signal for each source (e.g., even with very little data): if we have flexible compute, consider iteratively using more compute for the proxy models until some stability in the mixtures is reached”
>
> > Do the authors have any concrete ideas of how MixMin could be adapted in the future to handle data sources with covariate shift (i.e. image classification)? To be clear, I don’t think that concrete ideas for this are necessary to include in this paper, but I am intellectually curious.
>
> For covariate shift, the primary bottleneck we hit was non-convexity. The formula for the Bayes optimal function of a mixture with covariate shift is a fraction with a linear model of the mixture weights in the numerator and denominator (see Appendix A.1 formula). When we naively ran entropic descent to optimize the mixture with this expression we hit instability, and couldn’t get convergence without carefully tuning the learning rate. Even then it was not clear to us if this was a global optima. In fact, in preliminary experiments directly optimizing the conditional p(A|Q) tasks for LLMs  (which was non-convex given covariate shifts in the question strings) gave worse mixtures to our approach of making the task generative, i.e., learning p(Q,A).
>
> An analog of our approach, replacing p(y|x) tasks with p(y,x), could be done for image classification. People are currently training generative models to jointly learn the distribution of labels and images (multi-modal models). In this case MixMin (with no further modifications) can be used despite there being “covariate shifts” between the sources (i.e., image shifts).

---

> > ### Comment · Reviewer_KxeH · 2025-04-01
> >
> > Thank you for answering all of my questions. I agree with the author's proposal of adding this provided list of "weak conditions for when MixMin works" to the paper. I maintain that this work should be accepted.

---

### Decision · Program_Chairs · 2025-05-01

**Decision:**

Accept (poster)

**Comment:**

This paper proposes a method for optimizing the proportions of data that should be used from multiple sources in training large machine learning models. The idea is to train inexpensive proxy models on each data source individually, and then leverage the performance of these proxy models on downstream tasks to reason about the value of the individual data sources.

The reviewers highlight the empirical results and relevant topic tackled by the paper, and appear to have been satisfied with the author feedback responding to any questions, which largely amounted to minor requests for additional clarity.